# Development of a Set of Synthetic Diagnostics for the Confrontation between 2D Transport Simulations and WEST Tokamak Experimental Data

Ivan Kudashev [1,*], Anna Medvedeva [1], Manuel Scotto d'Abusco [1,2], Nicolas Fedorszak [2], Stefano Di Genova [1], Vladislav Neverov [3] and Eric Serre [1]

1    M2P2 Laboratory, CNRS Centrale Marseille, Aix-Marseille University, 13013 Marseille, France
2    IRFM, CEA Cadarache, 13108 Saint Paul-lez-Durance, France
3    National Research Centre 'Kurchatov Institute', 123182 Moscow, Russia
*    Correspondence: ivan.kudashev@univ-amu.fr

**Featured Application: The results of this study could be applied for SolEdge3X-HDG code validation with respect to experiment measurements as well as for improvement of the existing diagnostics on the WEST tokamak.**

**Abstract:** Transport codes are frequently used for describing fusion plasmas with the aim to prepare tokamak operations. Considering novel codes, such as SolEdge3X-HDG, synthetic diagnostics are a common technique used to validate new models and confront them with experimental data. The purpose of this study is to develop a set of synthetic diagnostics, starting from bolometer and visible cameras for the WEST tokamak, in order to compare the code results with the experimental data. This research is done in the framework of Raysect and Cherab Python libraries. This allows us to process various synthetic diagnostics in the same fashion in terms of 3D ray tracing with volume emitters developed specifically for fusion plasmas. We were able to implement the WEST tokamak model and the design of bolometer and visible cameras. Synthetic signals, based on full-discharge WEST plasma simulation, were used for to compare the SolEdge3X-HDG output plasma with experimental data. The study also considers the optical properties of the plasma-facing components (PFCs) and their influence on the performance of diagnostics. The paper shows a unified approach to synthetic diagnostic design, which will be further extended to cover the remaining diagnostics on the WEST tokamak.

**Keywords:** fusion; synthetic diagnostics; bolometry; visible camera; transport code; Raysect; Cherab





## 1. Introduction

To optimize the plasma parameters and the design of scenarios for tokamak operations, significant scientific efforts have been focused on experimental and theoretical studies of the tokamak power exhaust. One of the main challenges in this domain lays in the harsh tokamak environment for direct measurements and in simplifying the hypotheses made for the numerical simulations of plasma. In this context, synthetic diagnostics are the main tool able to confront plasma transport codes and experimental data from the full set of plasma diagnostics.

Numerous studies have been conducted in the field of plasma modeling for the description, prediction, and understanding of the evolution of plasma. The most common 2D transport codes use field-aligned meshes, as in SolEdge2D-Eirene [1] and SOLPS [2]. Despite the remarkable level of development, this approach still has several problems. First of all, it is not a straightforward way to describe the complex details of the tokamak vessel and plasma-facing compoments (PFCs). To deal with this issue, 3D SOL codes using Monte Carlo methods can be employed, such as in EMC3-Eirene [3]. In addition, singularities

occur in the vicinity of X-points of the magnetic field and, more importantly, in the center of the plasma column. Therefore, it is not possible to perform simulations for the whole plasma domain, and such codes as SOLPS do not consider the center region at all. The key plasma parameters at the last closed magnetic flux surface are matched with other simulations, for example, by using the ASTRA [4] simulation. Moreover, if there is a need to change the magnetic field configuration, one should redefine the mesh, which is extremely time-consuming. Consequently, it is almost impossible to perform 2D simulations for full discharges with evolving magnetic equilibrium.

Recently, a novel approach was introduced for fusion plasma modeling by using the hybridized discontinous Galerkin (HDG) method [5]. It employs non-aligned, non-structured meshes which allow us to not only precisely describe the PFCs geometry and the evolving magnetic field, but also to refine some mesh domains locally. This approach provides a better plasma description together with good time performance. The application of the HDG code led to the state-of-the-art 2D simulation of the entire discharge in the WEST tokamak from the plasma start-up to its ramp-down [6]. It also demonstrated the difference of such an approach compared with the usual steady-state ones. The plasma density in the latter case is shown to be lower than in a full discharge simulation, which cannot be covered by the commonly used codes. However, despite the promising qualitative agreement of the evolving plasma parameters with the experimental values, quantitative comparison with the integrated electron density from interferometry measurements shows discrepancies [6]. This means that the HDG code still needs more improvements and investigations, as well as benchmarking with both experiments and other existing codes.

The tokamak environment with its high temperatures (of the order of $10^7$–$10^8$ K) and low pressures (few Pa) usually leads to indirect measurements of macroscopic plasma parameters, which are typically compared with the outputs of the transport codes. Each measurement then needs to be interpreted, applying a set of assumptions in the physical model of a diagnostic and often using hypotheses on the plasma state and its equillibrium. For example, the phase shift of a passing laser beam can stand for the line-integrated density, supposing that electron temperature does not affect the measurement, or the intensity and the wavelength spectrum of the light can give information about density and temperature of radiating plasma species if we assume a certain model for the ionization states distribution [7], etc. Synthetic diagnostics are used to verify such relations. In addition to the evaluation of the fidelity of the imposed simplifications, it can also be employed for the benchmarking of the transport codes.

One of the widest family of the tokamak diagnostics are the ones dealing with plasma radiation. The Cherab [8] Python library, which is based on the Raysect [9] ray-tracing framework, was designed specifically to describe plasma radiation diagnostic systems. Raysect is a 3D spectral geometric ray tracer, which does not cover dispersion and dissipative effects. It has a collection of observers and, so-called primitives ranging from simple geometric shapes to detailed meshes for sophisticated computer-aided design (CAD) models. The user can specify surface and volume properties of the primitives such as bidirectional reflectance distribution function (BRDF) or anisotropic volumetric radiation. With the help of the Cherab library, various plasma parameters, such as magnetic configuration or particle distributions can be associated with the primitives. The more detailed description of the libraries with the underlying equations can be found in [8]. All in all, the Cherab–Raysect framework allows us to comprehensively describe both plasma radiation and optical properties of PFCs in the common fashion for different diagnostics. Here are only a few of its applications: bolometry on JET [10], charge exchange recombination spectroscopy (CXRS) diagnostics on COMPASS upgrade [11], and design of $H_\alpha$ emission diagnostics on ITER [12]. Moreover, SolEdge3X-HDG has the potential to cover the 3D plasma domain as well as the details of the tokamak PFCs. Even though toroidal symmetry is usually supposed, localised tokamak elements will cause the simulated plasma parameters to vary not only in poloidal cross-section, but also with toroidal angle, as in the real devices. Moreover,

visible diagnostics are influenced by reflections on the vessel elements. Hence, employing a 3D ray-tracing tool will lead to a more realistic description for the synthetic diagnostics.

The development of synthetic diagnostics using Cherab–Raysect framework is now ongoing for the WEST tokamak and is improving both the measurement quality and its interpretation. For example, bolometer cameras are used for radiated power measurement. On the one hand, this diagnostic can be used to evaluate the power radiated by heavy impurities in the core, which is not favorable for fusion reaction. On the other hand, it can be used to control the particle heat flux on the divertor plates by following the light impurity injection. Although the bolometry system is well-designed and operational on the WEST tokamak [13], the synthetic diagnostic code SYNDI [13] currently used to predict bolometry signals is not able to handle non-axysimmetric geometries of PFCs and the tokamak vessel. Moreover, because bidirectional BRDF is still under investigation for the WEST PFCs [14], one can use the Cherab library to adjust optical properties in the model by using a similar approach to that used in [8] and investigating the influence of different materials on the signals.

Another form of diagnostics which are still not widely used on the WEST tokamak are visible cameras, which are now operated for observation purposes. This instrument could be a powerful tool for numerous tasks such as plasma boundary or last-closed flux surface (LCFS) detection [15,16], attention-needed area detection and localization (which also can employ neural networks) [17,18], or even for tomographic inversion of detected radiation [19] and calculation of the sources of particles from these. However, visible cameras are struggling with the poor description of the reflection properties of the PFCs [8], which can lead to misinterpreting of the experimental images. Cherab and Raysect allow us to deal with this issue, introducing flexible ways to describe optical properties of the surfaces and to implement visible cameras' technical characteristics. Together with simulation in SolEdge3X-HDG, this can lead us to the design of novel, more comprehensive ways to exploit such diagnostics on the WEST tokamak.

All in all, novel codes, such as SolEdge3X-HDG, as well as the physical models of the experimental diagnostics are the subject of validation and benchmarking. This could be done in the framework of the Cherab and Raysect Python libraies. Moreover, the co-use of transport codes and synthetic diagnostics will be beneficial for the experimental diagnostics setup of the WEST tokamak. Therefore, in this paper we introduce the development of a Cherab machine-specific package for the WEST tokamak and SolEdge3X-HDG codes. It contains the description of the first synthetic diagnostics, namely for bolometry and visible camera. The benchmark between the existing WEST ray-tracing code SYNDI, experimental signals, and the developed synthetic one is provided. The influence of the roughness of the material and reflections model on the simulated signals is also shown. Finally, the ability to simulate visible camera images for different phases of the plasma discharge is demonstrated.

We organize this paper as follows. In Section 2, we briefly discuss the SolEdge3X-HDG and ERO2.0 code, the Cherab–Raysect framework and describe the discharge parameters used for simulation. Moreover, the WEST bolometer system is introduced. Section 3 is dedicated to the performance of the synthetic diagnostics, including bolometer system benchmarking and demonstrating the images for the visible camera. We discuss our results and show possible further extensions of the work in Section 4.

## 2. Synthetic Diagnostics and Simulation Data

The data used for generation of the synthetic diagnostic signals was taken from Ref. [6] and obtained by SolEdge3X-HDG simulation. The model is based on the Braginskii conservative equations for density, parallel momentum, total energy for singly charged ions, and electrons for the entire plasma volume. Neutral dynamics by simple diffusion model as well as Ohmic heating are also considered in the code. The latter one is calculated based on the current profile, obtained from the experimental data. Here, we will briefly

describe only the simulation parameters. For a more detailed description of the code, the reader can refer to [5,6].

The WEST discharge #54487 was chosen for simulation of the diagnostics in this paper and the results are taken from [6]. There are 403 different time steps with an interval of $dt = 0.02$ s. Perpendicular diffusion coefficients $D$, $\mu$, $\chi_i$, $\chi_e$ (for particle, momentum and ion and electron energy transport, respectively, across the magnetic field) are constant and equal to 0.5 m²s⁻¹. Parallel diffusion coefficients for ion and electron energy transport along the magnetic field lines are equal to $k_{\|,i} = 60$ [Wm⁻¹V⁻⁷/²], $k_{\|,e} = 2000$ [Wm⁻¹V⁻⁷/²]. The neutrals diffusion coefficient $D_{n_n}$ is set to 2000 m²s⁻¹. The values of the transport coefficients were adjusted in a way to achieve the best possible match between line-integrated density in simulation with the interferometer data. Neutral sources are defined by pumping rates and recycling. The former was obtained from the WEST experimental database. The Eirene code was not used in this simulation in contrast to [1]. The recycling coefficient $R$ is equal to 0.998. The mesh used in the SolEdge3X-HDG as well as in Cherab is shown in Figure 1.

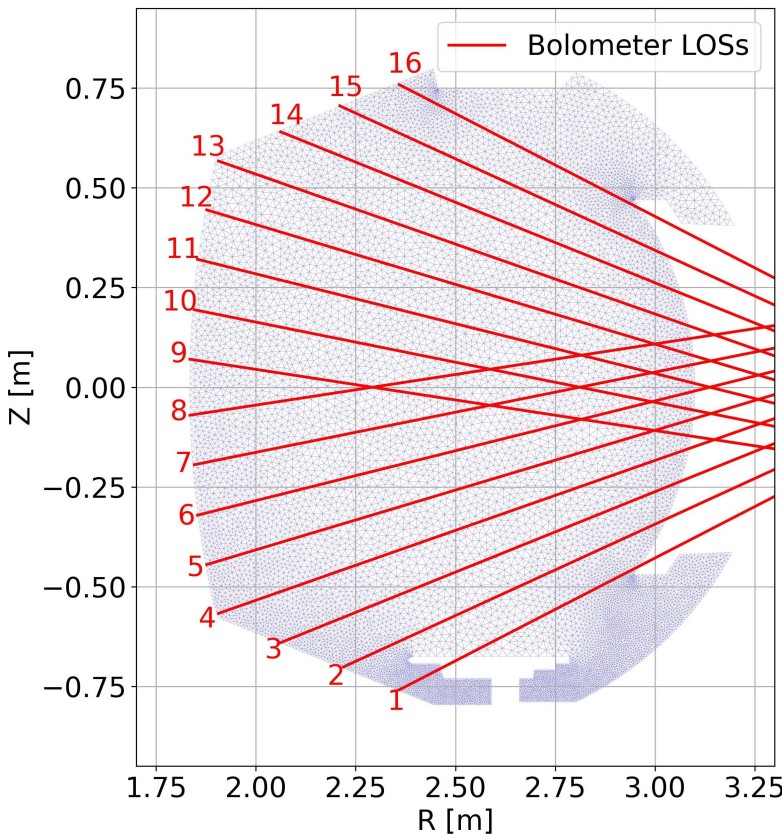

**Figure 1.** SolEdge3X-HDG simulation mesh and lines of sight (LOSs) of bolometer system of the WEST tokamak, which is located in the horizontal diagnostics port.

In addition, tungsten erosion was calculated with the Monte Carlo code ERO2.0 [20,21] with a SolEdge3X-HDG output as a plasma background. The latter one provides spatial distribution of $n_e$ (equal to the ion density), $T_e$, the main ion temperature $T_i$, the main ion parallel velocity $v_\|$ and the magnetic field **B**. ERO2.0 is a 3D code; however, the discussed simulation was 2D, and therefore the wall geometry was just an axial extrusion of 2D poloidal section used in SolEdge3X-HDG. Such an assumption leads to the antenna-limiter being axysimmetric rather than a set of toroidally distributed antennae, as in the experiment. As was discussed in [22,23], this implies a slight shift of the antenna off the plasma. Therefore, the erosion of tungsten is less compared with using the real 3D geometry of these PFCs [23]. This might be one of the reasons of bolometer signal

descrepancy, as will be discussed in Section 3.3. Oxygen is used as a proxy for light impurities in the ERO2.0 simulation, which are the main source for the tungsten sputtering. Its concentration was set to be constant at 3% over the plasma domain with fractional abundances, corresponding to [24] at the targets ($n_{O^+} : n_{O^{2+}} : n_{O^{3+}} : n_{O^{4+}} : n_{O^{5+}} : n_{O^{6+}} : n_{O^{7+}} : n_{O^{8+}} = 0.4:0.15:0.15:0.07:0.1:0.015:0.015$). The more precise ERO2.0 model description can be found in [22,23]. The output of the code provide us with a map of tungsten densities $n_{W^{Z+}}$, where Z corresponds to the charge of the ion (or 0 in case of the atom).

Obtained from SolEdge3X-HDG maps of the $n_e$, $n_{D^+}$, $n_{D^0}$, $T_e$, $T_i$ (equal for all of the neutrals and ions), and $n_{W^{Z+}}$ from ERO2.0, are used to reproduce plasma radiation by using Cherab and Raysect. A constant concentration of oxygen of 3% as in ERO2.0 simulation is used as a light impurity proxy. Fractional abundances are calculated with ionisation, recombination, and thermal charge exchange rate coefficients. The parameter grid and interpolation on the HDG values are rough, so oxygen concentration is employed only to show qualitative sensitivity of the method to the presence of light impurities. Four time moments, corresponding to the limiter, ramp-up, flat-top, and ramp-down stages with $t = 0.26, 1, 4.48$, and $7.52$ s are chosen. The mentioned maps are interpolated by using built-in functions, and the 2D profiles on corresponding meshes are shown in Figures 2 and 3. Plasma is modeled as an axysimmetric volume emitter. OpenADAS [25] is used for atomic data.

To calculate the bolometer signals, the TotalRadiatedPower model from Cherab is applied. It uses the ADF11 subpackage of ADAS, PLT files for line excitation radiation, and PRB for continuum and line recombination and bremsstrahlung power losses. Moreover, the bolometer, bolometer slit and foils classes from Cherab are employed to describe the system. The horizontal WEST tokamak bolometer system with two cameras, each with eight fan-spanned channels, which was operated in previous campaigns, is used in this article. Each of the two horizontal cameras contains of slits of dimension $9 \times 3.4$ mm positioned at $R = 4.482$ m, $Z = \pm 0.336$ m, and toroidal angle $\phi = 84.88°$. Ten centimeters behind the slit, eight bolometer foils of dimension $3.8 \times 1.3$ mm are positioned vertically every 5 mm. The LOS are shown in Figure 1. The error, associated with the experimental data are in the range from 1 to a few %, linked to calibration procedures and subtraction of low-frequency integration drift [26].

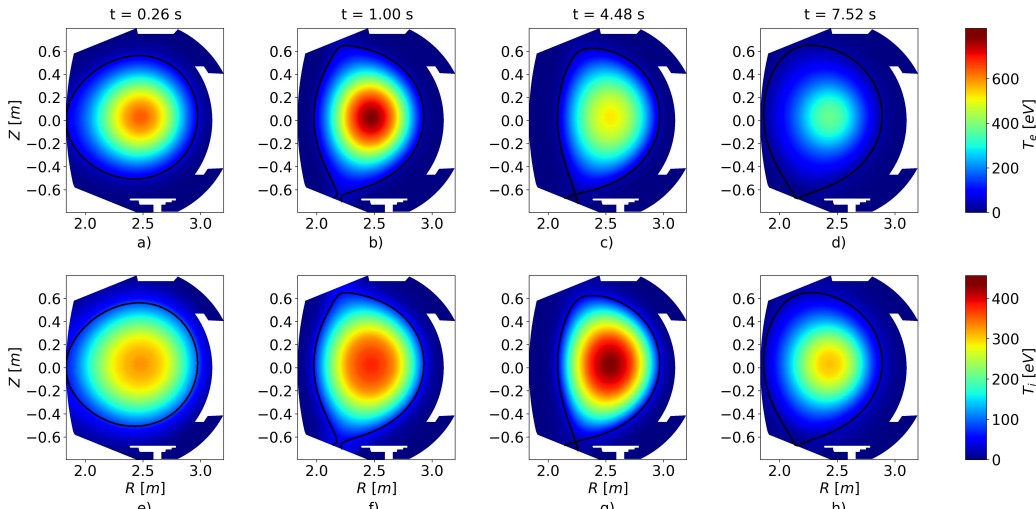

**Figure 2.** Time evolution of electron $T_e$ (**a–d**) and ion $T_i$ (**e–h**) temperature isolines on snapshots for $t = 0.26, 1, 4.48, 7.52$ s. Separatrix is shown by the black solid line.

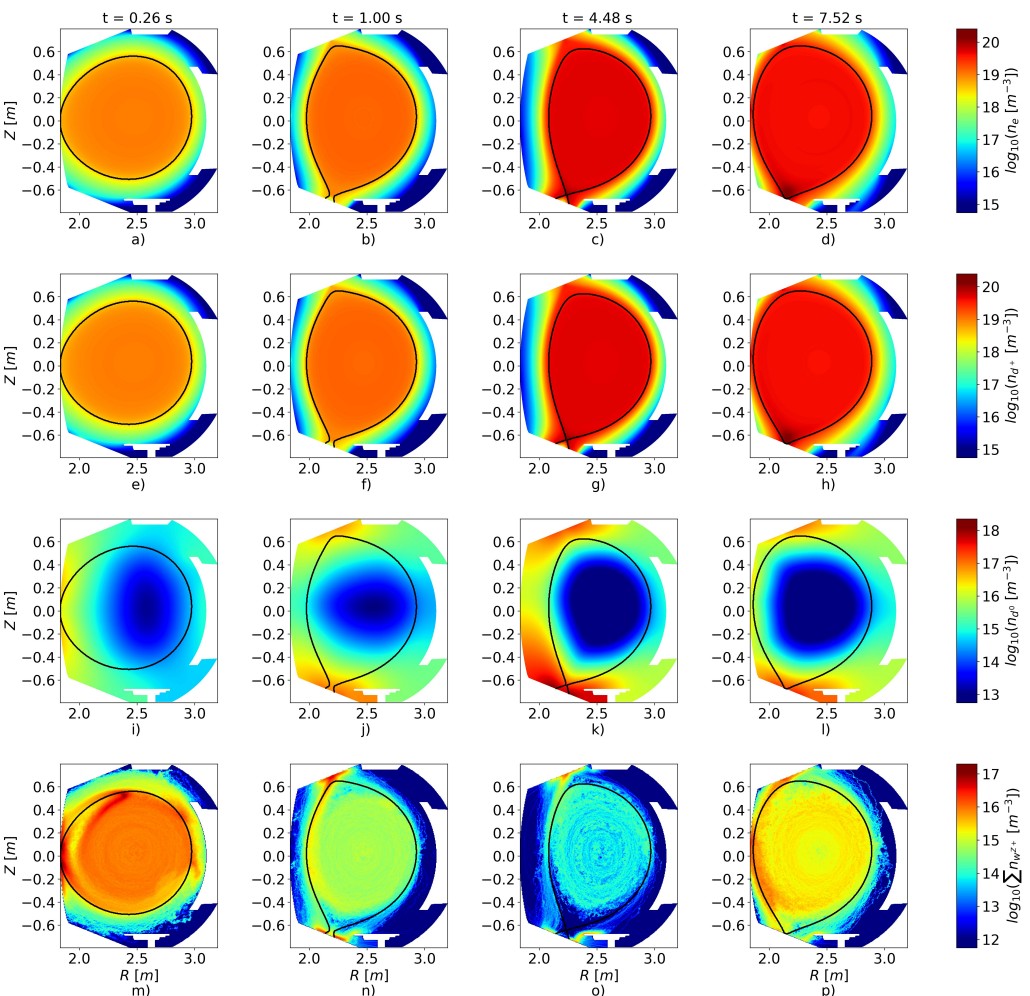

**Figure 3.** Time evolution of isolines of logarithms of electron $n_e$ (**a**–**d**), deuterium ions $n_{D^+}$ (**e**–**h**), deuterium neutrals, $n_{D^0}$ (**i**–**l**) and total tungsten $\sum n_{W^{Z+}}$ (**m**–**p**) densities on snapshots for $t = 0.26, 1, 4.48, 7.52$ s. Separatrix is shown by the black solid line.

The PinholeCamera Raysect class with $512 \times 512$ pixels was used to introduce the availability and performance of the synthetic visible camera. It is positioned at equatorial plane $R = 3.085$ m, $Z = -0.2$ m, $\phi = -139.6°$, and $20°$ from the Y-axis in the clockwise direction. The field of view was set to $45°$. For visible camera image generation WI 401.22 nm, deuterium Balmer series recombination and excitation lines, as well as bremsstrahlung are used.

Simplified models of the WEST vessel and PFCs were obtained from the ToFu library [27]. As this vessel model lacks port descriptions, the walls were omitted, and only PFCs were taken into account while calculating the bolometer signal. To avoid double counting of the plasma signal for central channels (when a ray can pass through plasma twice due to the absence of the wall) an absorbing proxy cylinder was placed in the center of the scene. At this stage of the work, we assume all of the WEST components to be made of the same material. Where the material is not mentioned, an absorbing surface was used. This approach provides enough accuracy for the initial benchmark and does not require much effort. However, in the dedicated Section 3.2, we compare several methods of PFC description, which include the JET description for bulk tungsten and for PFCs, coated with tungsten [8] and a specular tungsten. These assumptions, probably being not exactly correct, still cover a significant range of possible optical properties and, therefore, they influence the diagnostics.

## 3. Results

### 3.1. Bolometer Benchmark with SYNDI

First of all, the Cherab model of the bolometer cameras and WEST PFCs models should be benchmarked with the existing software for the WEST tokamak bolometers, SYNDI. The radiation profile, corresponding to $t = 4.48$ s, the flat-top, and the lower divertor stage of the discharge, are obtained from the simulated plasma background (Figure 3c,g,k,o), taking into account deuterium and tungsten. Bolometer signals are calculated by using the Cherab–WEST package and SYNDI software. Comparison of the synthetic signals as well as the radiation profile itself is shown in Figure 4.

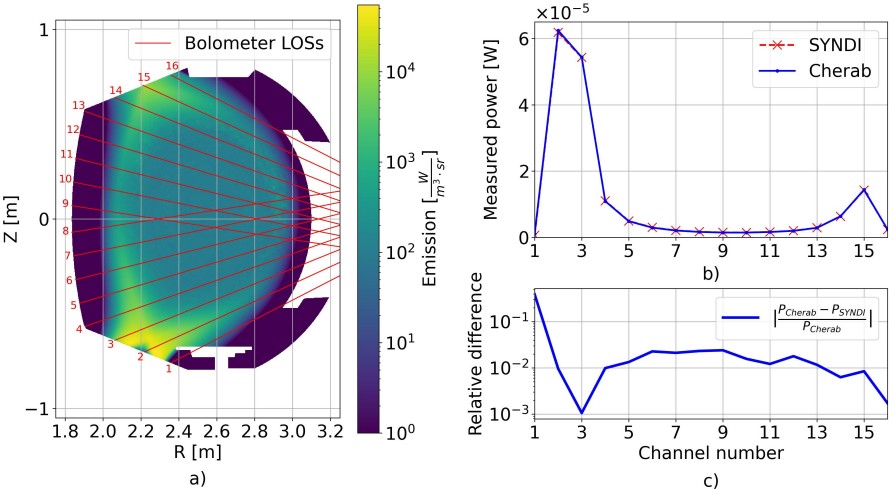

**Figure 4.** Bolometer cameras benchmark. (**a**) Two-dimensional radiation isolines, obtained with Cherab and used for signals calculation. (**b**) Synthetic signals from Cherab and SYNDI comparison. (**c**) Absolute relative difference between Cherab and SYNDI calculated signals.

The results of Cherab and SYNDI bolometer signal calculation agree with each other almost perfectly (Figure 4b,c), except for the first channel. This might be caused by the very low radiation measured by this channel. Such a discrepancy does not seem to be of great importance; however, it should be taken into account in further studies. In addition, Cherab could be a useful tool to also check whether or not the LOSs are shadowed by the tokamak parts. This opportunity may be used to validate exact position of the cameras on a real device.

### 3.2. Different Surfaces

One of the crucial parameters for signal simulation is the proper surface choice. Here, we compare fully absorbing, lambertian, rough (with roughness of 0.29 in the Cook–Torrence BRDF), and specular tungsten surfaces. The latter three models have 10%, 45%, and 54% of effective reflectivity averaged over the angles of incidence. The absorbing and specular cases give the most extreme cases setting the range of possible influence of the reflections. The lambertian and rough models are supposed to describe tungsten-coated and bulk tungsten PFCs on the JET tokamak, respectively [8]. As most of the WEST PFCs were coated with tungsten, but not made from bulk tungsten, the lambertian case might be the closest to the real optical properties of WEST components.

To obtain data from Figure 5, deuterium and tungsten radiation is taken into account. Time moments of $t = 0.26$ and $4.48$ s are used as the most representative ones, covering both limiter and divertor phases of the discharge. According to Figure 5, the signals diverge both for divertor and central channels. This is due to the complex geometry of the PFCs, which allows reflected light from various locations to reach the detectors. Such a difference can be important both for a simulation results benchmark as well as for the interpretation of the experimental results. For example, this is crucial for tomography inversions, because

reflected light can be counted as the real plasma radiation. The proper optical description of the WEST tokamak should be implemented during further studies together with a more precise CAD model of the WEST tokamak.

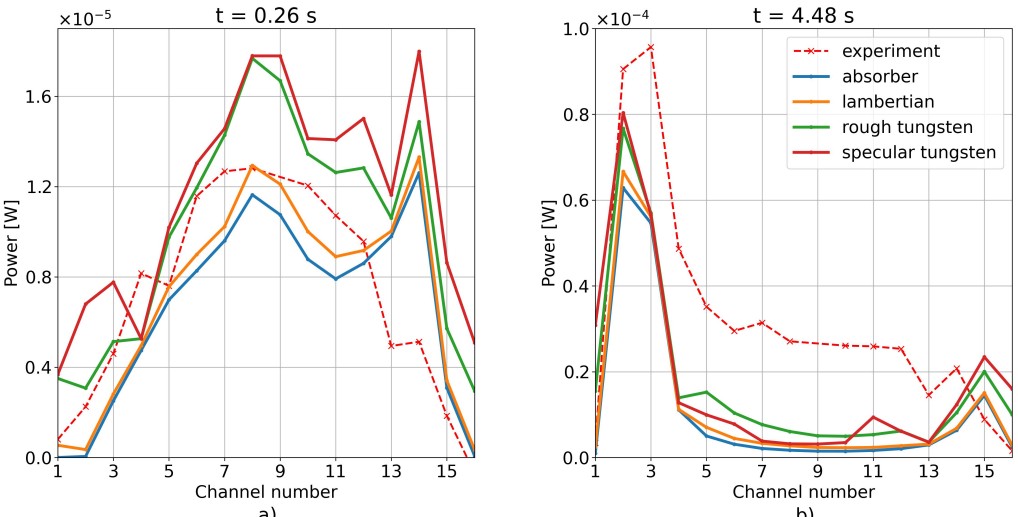

**Figure 5.** Influence of the different optical description of the WEST PFCs on the bolometer signals for absorbing, lambertian, rough tungsten, and specular tungsten surface models for limiter (**a**) and divertor (**b**) phases.

### 3.3. Confrontation between Simulated and Experimental Bolometer Signals

To compare with the experiment, four stages of the discharge were chosen, first corresponding to the limiter phase and the other three to the divertor phase: ramp-up, flat-top, and ramp-down. These correspond to times of $t$ = 0.26, 1, 4.48, and 7.52 s. The experimental channel #9 was affected by some calibration issues, so it is omitted from the plots. We demonstrate individual contributions from deuterium and tungsten, as well as from proxy oxygen to assess the sensitivity of the Cherab framework to the different sources of radiation. The comparison of the simulated and experimental results is demonstrated in Figure 6.

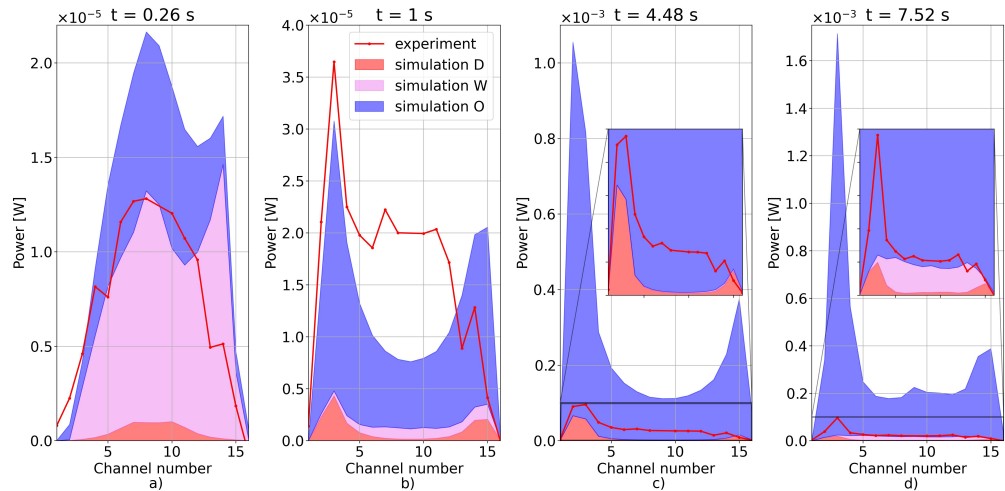

**Figure 6.** Comparison between bolometer experimental and simulated signals for $t$ = 0.26 (**a**) (limiter phase), 1 (**b**), 4.48 (**c**), and 7.52 s (**d**) (divertor phase). Contributions from deuterium (orange), tungsten (magenta), and oxygen (blue) are shown on the simulated signal.

We can see a good agreement of deuterium and tungsten contributions with experimental signals for the limiter phase of the discharge. During this stage, sputtering occurs

from the HFS limiters, which can be seen in Figure 3. However, there is a signal spike for the channels from 12 to 14 (Figure 6a), which is caused by a tungsten radiation contribution. According to Figure 3m, tungsten is accumulated near the separatrix at the upper part of the plasma. This is a rather numerical artifact, which probably originates from the fact that SolEdge3X-HDG is not field-aligned. Therefore, when passing output data to the ERO2.0 input format, calculated parallel temperature gradients might be too high. This leads to higher thermal forces tending to extra tungsten accumulation. Despite the fact that this effect is a theoretically predicted effect [28], it should be less pronounced. This numerical error will be eliminated improving the compatibility of SolEdge3X-HDG and ERO2.0 codes.

The total synthetic signals, including oxygen contribution, is of the same order as the experimental signal for $t = 1$ s. However, we obtain a huge overestimation of oxygen radiation at the later phases of the discharge. This is due to the fact that our simplified model of constant oxygen concentration is not good enough to describe experiment. Moreover, oxygen concentration might be changing in time during the discharge. To have a good comparison with the experiment for each time moment a scan over different oxygen concentrations both for ERO2.0, as well as for Cherab, should be made. However, it was not the goal of this paper. At the $t = 1$ and 4.48 s, there might also be a tungsten erosion underestimation; at these stages, LFS limiters are supposed to be under high particle flux, but, as was mentioned before, in current ERO2.0 simulation these PFCs were moved slightly away from the plasma. In addition, according to the experimental data at $t = 4.48$ s, the LFS limiter was close to the separatrix, which is not taken into account. This may cause an increase of radiation in the experiment, which is not covered by simulations. There is one more possible reason of discrepancy, the diffusion coefficients are taken constant, which is a simplification. The resulting electron and ion profiles from SolEdge3X-HDG simulation also affect ERO2.0 calculations. It is well known that due to neoclassical effects, impurities tend to peak in the core region [29]. These processes, as well as turbulence transport, should be taken into account in further versions of the SolEdge3X-HDG for more accurate simulations.

Nonetheless, from such a comparison we can clearly see the sensitivity of Cherab simulations to the presence of different radtiation sources. Together with the mentioned possible reasons of the discrepancies, we can use the WEST package to assist physical model improvement.

### 3.4. Visible Camera

Here, we demonstrate the performance of a visible camera by using Cherab in Figure 7. A simple pinhole model with $512 \times 512$ CCD pixels was used. No optical elements, such as lenses, were modeled. The exposure was adjusted manually to have appropriate image brightness. Moreover, the position and observation geometry used in this study reproduce the real camera only approximately. These will be improved in further studies with the use of calibration tools, for example, Calcam [30].

Because two different radiation patterns are usually observed during limiter and divertor discharge state, we chose $t = 0.26$ and 4.48 s to simulate images. We use bremsstrahlung radiation, deuterium Balmer lines ($\alpha$, $\beta$, $\gamma$, $\delta$ and $\epsilon$), and a WI 401.22 nm line for visible camera and a single WI 401.22 nm line for the filtered camera simulations. The rough tungsten model from Section 3.2 was used for the description of WEST PFCs. From the comparison of the visible experimental camera (Figure 7c,d) with simulated images (Figure 7a,b), similar patterns can be distinguished, such as brighter light along the separatrix, the X-point, and in the divertor region. Moreover, different radiation patterns correspond to different discharge stages, so, for example, a visible camera can be used to define whether the plasma detached or not. Furthermore, adding more impurities, which radiate more in the violet region, might also help match the color of the images.

It is clearly seen that more bright parts of the filtered pictures in Figure 7e,f correspond to the points, where the magnetic field crosses the material surfaces: high-field side limiter and divertors for limiter and divertor stages, respectively. The concentration of impurities

can be obtained from such images. Therefore, a similar digital twin, a virtual representation of a real-world system, i.e., a numerical description of camera combined with plasma simulation, may be used for the diagnostic design.

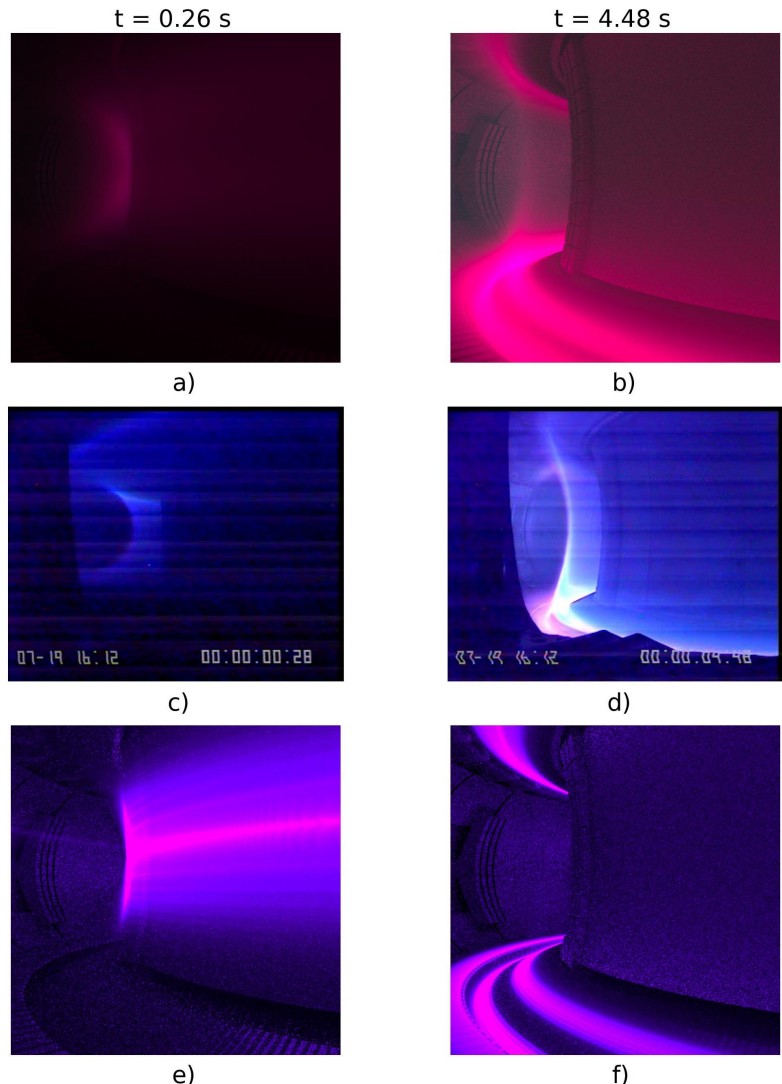

**Figure 7.** Simulated images for visible pinhole camera for limiter (**a**) and divertor flat-top (**b**) phases of the discharge, experimental visible camera images (**c**,**d**) and a filtered pinhole camera for the WI 401.22 nm line (**e**,**f**) for the similar discharge stages.

## 4. Discussion

In this paper, we demonstrated the first results of synthetic diagnostics Cherab package development for usage with SolEdge3X-HDG code and WEST tokamak. It includes bolometer and visible camera simulated signals as well as a comparison of the 2D transport code results with experimental data.

At first, the bolometer submodule was verified with an existing ray-tracing code SYNDI. Even with a simplified tokamak PFCs model in the absence of reflections, the comparison is almost perfect. This means that the geometry set in both codes, as well as line integration, are identical. However, later, a more detailed WEST CAD model will be used, and this will give more relevant synthetic signals compared to the experimental ones. Moreover, when the new vertical bolometer systems are commissioned, Cherab–WEST package might be used to check the geometry and relative position of both cameras and tokamak elements.

Another useful implementation of these WEST digital twin will be a proper selection of surface material model. As was shown in Section 3.2, there might be a difference of order of magnitude in the case of different optical PFC properties. Not only the value of the modeled signal, but also the shape of the bolometer signals profile can change significantly. This may cause a systematic error while interpreting experimental results in terms of the locations of the highest radiation. Together with the BRDF measurements [14], a better refined CAD model, and better Cherab and Raysect libraries, a physically reliable WEST tokamak model should be implemented.

In comparison with the experimental data, using deuterium, tungsten, and proxy oxygen showed the sensitivity of our bolometer digital twin to the different sources of radiation. However, it also revealed discrepancies, which were caused by the model's simplicity and uncertainties. Nonetheless, having such a confrontation tool will allow us to more easily improve the model in SolEdge3X-HDG, as well as in ERO2.0.

In addition, the very availability of a full-discharge, entire-plasma-domain simulation with SolEdge3X-HDG allows us to perform benchmarking for the diagnostics which use line-integrated signals, i.e., cover the core plasma. Consequently, there is a possible co-use of synthetic diagnostics and this novel code dedicated to improvement of the existing experimental tools. For example, bolometers are not yet able to perform real-time tomography measurements during discharge. For these purposes, machine learning can be used [31,32]. At the same time, SolEdge3X-HDG could be utilized to obtain the training set as the vertical bolometer system was not in use before. Therefore, there is no opportunity to get the 2D radiation profiles from only horizontal systems without strict assumptions on its distribution. Profiles obtained from simulation have another advantage over experimental ones; they are less noisy, and the code can cover more regimes in less time than in desired experiments. Among these applications, Cherab can be also employed for performing the conventional tomography inversions, which also should be implemented for the upcoming campaign. Moreover, it is a very straightforward tool with which to obtain geometry matrices for further usage with other software, for example, ToFu [27], which is designed for tomography inversions.

The filtered visible camera demonstration shows the ability to use such a device in the real experiments. In the easiest case, it can show the approximate location of the particle sources. However, it is possible to make a tomographic inversion into the 2D radiation profile [8], which can further be interpreted in terms of particle fluxes. Moreover, the limiter and divertor phases are clearly distinguished on the visible images, as well as locations of the highest radiation. This might also be employed in the discharge state control system. On the top of that, visible cameras can be a useful tool for the LCFS detection, which was not investigated yet on the WEST tokamak.

Among the mentioned applications, the family of synthetic diagnostics will be expanded by at least visible spectroscopy diagnostics. Having digital twins of refractometry, soft-xray diagnostics, Langmuire probes, and interferometers, one will have a powerful tool with which to make a comprehensive confrontation of the transport code with the experimental data.

**Author Contributions:** Conceptualization, A.M. and E.S.; methodology, A.M., N.F. and I.K.; software, I.K., M.S.d., N.F., S.D.G. and V.N.; validation, I.K. and N.F.; analysis, I.K.; investigation, I.K.; resources, E.S.; data curation, I.K., M.S.d., N.F. and S.D.G.; writing—original draft preparation, I.K.; writing—review and editing, A.M., E.S. and N.F.; visualization, I.K.; supervision, E.S. and A.M.; project administration, A.M. and E.S.; funding acquisition, E.S. All authors have read and agreed to the published version of the manuscript.

**Funding:** This work has been supported by the French National Research Agency grant SISTEM (ANR-19-CE46-0005-03). This work has been carried out within the framework of the EUROfusion Consortium, funded by the European Union via the Euratom Research and Training Programme (Grant Agreement No 101052200-EUROfusion). Views and opinions expressed are however those of the authors only and do not necessarily reflect those of the European Union or the European

Commission. Neither the European Union nor the European Commission can be held responsible for them.

**Institutional Review Board Statement:** Not applicable.

**Informed Consent Statement:** Not applicable.

**Data Availability Statement:** The data presented in this study are available on request from the corresponding author.

**Conflicts of Interest:** The authors declare no conflict of interest. The funders had no role in the design of the study; in the collection, analyses, or interpretation of data; in the writing of the manuscript, or in the decision to publish the results.

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
