# Peer review of "Development of a Set of Synthetic Diagnostics for the Confrontation between 2D Transport Simulations and WEST Tokamak Experimental Data"

_applsci, doi:10.3390/app12199807_

Round 1
Reviewer 1 Report
This paper describes the development of a synthetic bolometer and visible camera diagnostic for SolEdge3x-HDG plasma model of the WEST tokamak. The synthetic bolometer measurement is benchmarked against and shows good agreement with the SYNDI code. This paper goes on further to show the results of synthetic signals compared to the experimentally measured real signal for a bolometer and a visible camera.
A major piece that is missing from this paper is the physics equations that describe how the synthetic signals are calculated. The paper mentions the python libraries Cherab and Raysect but does not describe how they are used to model specific attributes of the diagnostics. Is Raysect a simple 3D graphics ray tracer, or is it modeling the dispersion of rays through the plasma? More details on how the synthetic diagnostic calculation is needed.
The perpendicular diffusion coefficients D and Chi play a strong role in deterring density and temperature profiles. How were these coefficient values chosen? The synthetic data shows many disagreements between the synthetic signal and the measured signal, especially in the core region. Additionally, transport modeling also needs to take into account fueling sources and pumping sinks. How were these modeled in SolEdge3X-HDG? Results of this paper talk about the changes in signal to the optical and material properties of the first wall but if the plasma parameters are not correct, the synthetic diagnostic won't model the system at all.
The first plot in figure 5 shows the bolometer signal for t=0.26 seconds. In figure 3.a the plots shows the plasma is pretty close to up-down symmetric. Channel 14 shows a significant spike in the simulated signal. Channel three which is symmetrically opposite to that channel does not show this same spike. This is also reflected in the experimental data which shows a roughly symmetric pattern. What is the cause of this spike in this synthetic channel?
Measured signals always have some error associated with them. The author should include an uncertainty interval on the measured bolometer signals.
Section 3.4 needs a better description of the synthetic visible camera. What are the details on how is this image generated?
Reviewer 2 Report
The manuscript is devoted to the development of synthetic diagnostics of the bolometer and visible cameras and their verification by comparing the results of theoretical simulations (which uses 2D transport simulation with the new SolEdge3X-HDG numerical code applied to the 2D SOL plasma and modeling of light reflection from the 3D first wall of vacuum chamber) and experimental data of the WEST tokamak. Such work is very important for the development of the tools necessary to control the operation of the tokamak reactor. The trend towards the creation of unified synthetic diagnostics using modern IT tools and advanced methods for tokamaks physics modeling should be supported by the publication of such articles in peer-reviewed journals. I recommend this article for publication in Appl. Sci. MDPI journal requesting minor revision of the manuscript.
Line 28
Replace SolEdge2D with SolEdge2D-Eirene, according to the title of ref. [1].
Line 31
“Also, singularities occurs in the vicinity of X-points of the magnetic field and, what is more important, in the center of the plasma column. “
Peripheral plasma codes such as SOLPS do not consider the center of the plasma at all, they need to match a key parameter such as the heat flux density at the last closed magnetic flux surface, for example with ASTRA code simulation. Thus, the meaning of the phrase is not clear.
Line 37
A few words about 3D SOL plasma modeling for 3D PFC is required (e.g., EMC3-EIRENE code)
Line 51
“The tokamak environment with its high temperatures (of the order of 10^7–10^8 K) and low pressures (few Pa) usually does not allow to measure directly the plasma parameters, which are also typically the outputs of the transport codes.”
What about, e.g., Thomson scattering, passive and active spectroscopy? This phrase should be corrected. For example, for Thomson scattering, the transition from the {R, Z} coordinate to the magnetic flux coordinate does require simulation of the plasma equilibrium code (rather than simulation of the transport code), but the measurement itself is fairly straightforward.
Line 120
It is not clear if data for neutrals treated with EIRENE are present in the simulation [4], which only claimed fluid simulation. It is worth emphasizing that the EIRENE code is not used in the present study, in contrast to ref. [1].
Line 141
Replace “find” with “found”
Line 143, figure 1 caption
“… horizontal bolometer system …”
Figure 1 shows a poloidal plane in coordinates {R, Z}. It is worth clarifying that “horizontal” refers to the location of the diagnostic port.
Lines 170-172
It's not clear what was omitted. I assume it was the entire tokamak vessel model, so only the PFC model remained. If so, this should be stated more clearly.
Line 170
Please reformulate this sentence. I propose: “Simplified models of the WEST vessel and PFCs were obtained from ToFu library”.
Line 187
Replace “is” with “are”
Line 198
Dot after "surface" needed
Line 207
It is not clear from Figure 5 that only the divertors are affected. The curves diverge almost everywhere in both limiter and divertor phases.
Line 216
Omit “for comparison” to avoid repetition
Line 249
“pinhole visible camera”
A pinhole camera is a kind of diagnostic tool that is very different from a video camera. Is it appropriate to use such a term? "Pinhole" is not used in most of the article and may be omitted here and elsewhere.
Line 253
Replace “visible and only tungsten” with “visible camera and only the tungsten”
Line 261-263
The sentence is hard to follow, I propose this wording:
We use bremsstrahlung radiation, deuterium Balmer lines (α, β, γ, δ and ϵ), WI 401.22 nm line for visible camera and a single WI 401.22 nm line for filtered camera simulations.
Lines 264, 277
“digital twin” must be explained
Line 265
General comment on Section 3.4. Fig. 7 clearly shows that the pinhole model poorly reproduces the observation geometry of a real camera, but the authors do not mention how they plan to solve this problem.
Line 272
Not surprisingly, both packages give the same results, especially in the absence of reflections. Both packages use the same atomic data, so here the authors simply verified that Cherab performs integration along the lines of sight correctly. An appropriate comment is required.
Line 277-278
Replace “surface material choice” with “selection of surface material model”.
The authors can only choose the model, not the material. Therefore I propose the following wording: Another useful implementation of these WEST digital twins will be a proper selection of a surface material model.
Line 283
Insert a coma after “libraries”
Reviewer 3 Report
The author presents frontier research about synthetic diagnostics on WEST, which valid the 2D transport numerical simulation and experimental database. This research work is essential to improve data interpretation and simulation study.
(1) You mentioned the 3D ray tracing in abstract, but I don't see statement in the manuscript to discuss the 3D. I would like to add one short paragraph to describe the importance of 3D ray tracing in your model.
(2) Please add reference of SYNDI code.
(3) Please define and label of red lines in Figure 1. Is that possible to optimize the layout of these 16 paths? Such as tilt angle changes.
(4) In Figure 4a, I would like to recommend to add raw data (maybe one channel or more) which is used for contour figure generation.
(5) In Figure 4b, it is very difficult to see the difference between SYNDI and Cherab. It might be good to say these two results agree with each other perfectly. If so, may I ask for a separate figure to list the difference between SYNDI and Cherab.
(6) Miss sub-plot number (a),(b),... in Figure 5 and 6.
(7) Would you please provide the reference about how does the experiment data calibration? It is important for amplitude/power measurement.
(8) I could see the large dynamic range of receiving power from 2e-6 W (Figure 6a) to 1e-4 W (Figure 6d). It seems the receiver has excellent signal to noise ratio for high quality raw data. May I ask the full dynamic range of detectable power?
(9) In the summary(line 311), the author said the visible camera can be used for LCFS detection. I agree with that, but need delicate diagnostic setup and data interpretation.
